# Learning to Query, Reason, and Answer Questions On Ambiguous Texts

**Xiaoxiao Guo**[*]
Computer Science & Engineering
University of Michigan
guoxiao@umich.edu

**Tim Klinger**[*]
IBM Watson Research
Yorktown Heights, NY
tklinger@us.ibm.com

**Clemens Rosenbaum**[*]
Computer Science
UMass Amherst
cgbr@cs.umass.edu

**Joseph P. Bigus, Murray Campbell, Ban Kawas,**
**Kartik Talamadupula, Gerald Tesauro**
IBM Watson Research
Yorktown Heights, NY
jbigus,mcam,bkawas,krtalamad,gtesauro@us.ibm.com

**Satinder Singh**
Computer Science & Engineering
University of Michigan
baveja@umich.edu

## Abstract

A key goal of research in conversational systems is to train an interactive agent to help a user with a task. Human conversation, however, is notoriously incomplete, ambiguous, and full of extraneous detail. To operate effectively, the agent must not only understand what was explicitly conveyed but also be able to reason in the presence of missing or unclear information. When unable to resolve ambiguities on its own, the agent must be able to ask the user for the necessary clarifications and incorporate the response in its reasoning. Motivated by this problem we introduce QRAQ (Query, Reason, and Answer Questions), a new synthetic domain, in which a User gives an Agent a short story and asks a challenge question. These problems are designed to test the reasoning and interaction capabilities of a learning-based Agent in a setting that requires multiple conversational turns. A good Agent should ask only non-deducible, relevant questions until it has enough information to correctly answer the User's question. We use standard and improved reinforcement learning based memory-network architectures to solve QRAQ problems in the difficult setting where the reward signal only tells the Agent if its final answer to the challenge question is correct or not. To provide an upper-bound to the RL results we also train the same architectures using supervised information that tells the Agent during training which variables to query and the answer to the challenge question. We evaluate our architectures on four QRAQ dataset types, and scale the complexity for each along multiple dimensions.

## 1 Introduction

In recent years, deep neural networks have demonstrated impressive performance on a variety of natural language tasks such as language modeling (Mikolov et al. (2010); Sutskever et al. (2011)), image captioning (Vinyals et al. (2015); Xu et al. (2015)), and machine translation (Sutskever et al. (2014); Cho et al. (2014); Bahdanau et al. (2015)). Encouraged by these results, machine learning researchers are now tackling a variety of even more challenging tasks such as reasoning and dialog. One such recent effort is the so-called "bAbI" problems of Weston et al. (2016). In these problems, the agent is presented with a short story and a challenge question that tests its ability to reason about the events in the story. The stories require the agent to learn unstated constraints, but are otherwise self-contained,

---

[*]The first three authors contributed equally.

requiring no interaction between the agent and the environment. A very recent extension of this work (Weston (2016)) adds interaction by allowing the agent to respond in various ways to a teacher's questions.

There has also been significant recent interest in learning task-oriented dialog systems such as by Bordes & Weston (2016); Dodge et al. (2016); Williams & Zweig (2016); Henderson et al. (2014); Young et al. (2013). Here the agent is trained to help a user complete a task such as finding a suitable restaurant or movie. These tasks are typically modeled as slot-filling problems in which the agent knows about "slots", or attributes relevant to the task, and must determine which of the required slot values have been provided, querying the user for the others. The reasoning required to decide on an action in these systems is primarily in determining which slot values the user has provided and which ones are required but still unknown to the agent. Realistic task-oriented dialog, however may require logical reasoning both to minimize irrelevant questions to the user and to focus the inquiry on questions most helpful to solving the user's task.

In this paper we introduce a new simulator that generates problems in a domain we call QRAQ (Query, Reason, and Answer Questions). In this domain the User provides a story and a challenge question to the agent, but with some of the entities replaced by variables. The introduction of variables, whose value may not be known, means that the agent must now learn additional challenging skills. First it must be able to decide whether it has enough information, in view of existing ambiguities, to answer the question. This requires reasoning about which variables can be deduced from other facts in the problem. Second, if the agent cannot answer the question by reasoning alone, it must learn to query the simulator for a variable value. To do this it must be able to infer which remaining variables are relevant to the question posed. The agent is penalized for asking about irrelevant or deducible variables. Since there may be several rounds of questioning and reasoning, these requirements bring the problem closer to task-oriented dialog and represent a significant increase in the difficulty of the challenge over the original bAbI ("supporting fact") problems. In another significant departure from previous work on reasoning, including the work on the bAbI problems, we focus on the more realistic and challenging reinforcement learning (RL) setting in which the training agent is only told at the end of the multi-turn interaction whether its answer to the challenge question is correct or not. For an upper bound comparison, we also present the results of supervised training, in which we tell the agent which variable to query at each turn, and what to answer at the end.

In summary, this paper presents two main contributions: (1) a novel domain, inspired by bAbI, but which additionally requires reasoning with incomplete information over multiple turns, and (2) a baseline as well as an improved RL-based memory-network architecture with empirical results on our datasets that explore the robustness of the agent's reasoning.

## 2 Related Work

Our work builds on aspects of many different lines of machine learning research for which it is impossible to do full justice in the space available. Most relevant is research on deep neural networks and reinforcement learning for reasoning in natural language domains – in particular, those which make use of synthetic data.

One line of work which inspires our own is the development of novel neural architectures which can achieve deeper "understanding" of text input, thereby enabling more sophisticated reasoning and inference from source materials. In Bordes et al. (2010) for example, the model must integrate world knowledge to learn to label each word in a text with its "concept" which subsumes disambiguation tasks such as pronoun disambiguation. This is similar in spirit to our sub-task of deducing the value of variables but lacks the challenge of answering a question using this information or querying the user for more information. We draw particular inspiration for QRAQ from the bAbI problems of Weston et al. (2016) which are simple, automatically generated natural language stories, along with a variety of questions which can test many aspects of reasoning over the contents of such stories. The dynamic memory networks of Kumar et al. (2016) use the same synthetic domain and include tasks for both part-of-speech classification and question answering, but employ two Gated Recurrent Units (GRUs) to perform inference. Our problems subsume the reasoning required for the

bAbI "supporting fact" tasks and add additional complexity in several ways. First, we allow ambiguous variables which require logical reasoning to deduce. Second, the question is not necessarily answerable with the information supplied and the agent must learn to decide if it has enough information to answer. Third, when the agent does not have the information it needs it must learn to query the user for a relevant fact and integrate the response into its reasoning. Such interactions may span multiple turns, essentially requiring a dialog.

There has been a lot of recent interest on the end-to-end training of dialog systems that are capable of generating a sensible response utterance at each turn, given the context of previous utterances in the dialog (Vinyals & Le (2015); Serban et al. (2016); Lowe et al. (2015); Kadlec et al. (2015); Shang et al. (2015)). Research on this topic tends to focus on large-scale training corpora such as movie subtitles, social media chats, or technical support logs. Because our problems are synthetic, our emphasis is not on the difficulties of understanding realistic language but rather on the mechanisms by which the reasoning and interaction process may be learned. For large corpora it is natural to use supervised training techniques where the Recurrent Neural Networks (RNNs) attempt to replicate the recorded human utterances. However, there are also approaches that envision training via reinforcement learning techniques, given a suitably defined reward function in the dialog (Wen et al. (2016); Su et al. (2015b;a)). We adopt this approach and similarly emphasize RL in this paper.

As described in the previous section, most of the reasoning emphasis in the slot-filling model of task-oriented dialogs is on understanding the user goal and which slot values have been given/which remain unfilled. By contrast, in QRAQ problems the emphasis is on inferring missing information if possible, and reasoning about what is important to ask, which can be much harder than evaluating which of the required slots are still unfilled. In more recent work on end-to-end learning of task-oriented dialog such as Bordes & Weston (2016); Dodge et al. (2016) this paradigm is extended to decompose the main task into smaller tasks each of which must be learned by the agent and composed to accomplish the main task. Williams & Zweig (2016) use an LSTM model that learns to interact with APIs on behalf of the user. Weston (2016) (bAbI-dialog) combines dialog and reasoning to explore how an agent can learn dialog when interacting with a teacher. The dataset is divided into 10 tasks, designed to test the ability of the agent to learn with different supervision regimes. The agent is asked a series of questions and given various forms of feedback according to the chosen scheme. For example: *imitating an expert*, *positive and negative feedback*, etc. Our focus is not on comparing supervision techniques, so instead we provide a numeric reward function for QRAQ problems that only assumes knowledge of when the agent's answer is right. In bAbI-dialog, the agent cannot ask questions, only answer them and receive guidance from the supervisor. The QRAQ problems by contrast allow the agent to query for the values of variables as well as to answer the challenge question. The information received must then be integrated into the reasoning process since the decision about what to query next (or answer) is dependent on the new state. In the bAbI-dialog problems the correct answer to a given question does not depend on previous questions and answers, though previous feedback can improve the agent's performance.

## 3 THE QRAQ DOMAIN

QRAQ problems have two actors, User and Agent. The User provides a short story, set in a domain similar to the HomeWorld domain of Weston et al. (2016); Narasimhan et al. (2015), and a *challenge question*. The stories are semantically coherent but may sometimes contain unknown variables, which the Agent may need to resolve to answer the question.[1]

Consider the simple QRAQ problem in Example 1, where the context for the problem is labeled C1, C2, etc.; the events are labeled E1, E2, etc.; the question is labeled Q; and the ground truth is labeled GT. Here the entities $v and $w are variables whose values are not

---

[1]Formally, the solution requires many unstated assumptions which the Agent must learn to correctly solve the problem. These include: domain constraints (e.g. a person can't be in two places at once), domain closure (the only entities in the world are those referenced explicitly in the story), closed world (the only facts true in the world are those given explicitly in the story), unique names (e.g. Emma != Hannah, kitchen != garden, etc.) and the frame assumption (the only things that change are those explicitly stated to change).

provided to the Agent. In Event E3, for example, $v refers to either Hannah or Emma, but the Agent can't tell which. In a realistic text this might occur due to spelling or transcription errors, or the use of indefinite pronouns such as "somebody". Variables such as $u (which aliases Emma) might realistically occur as descriptive references such as "John's sibling". The question to be answered "Where is the gift?" is taken to mean "Where is the gift at the end of the story?", a qualification which the Agent must learn. We call the entity referenced in the question (in this case "the gift") the *protagonist*.

The variables occurring in this example can be categorized as follows: A variable whose value can be determined by logical reasoning is called *deducible*; a variable which is consistently used to refer to an entity throughout the story is called an *alias*; a variable whose value is potentially required to solve the problem is called *relevant* (non-relevant variables are called *irrelevant*). Aliased variables may be defined in the context (e.g. $u is Emma).

In Example 1, $x is irrelevant, since knowing its value doesn't help solve the problem. Similarly, we can observe that Emma (aliased as $u), leaves the garden in E5, making Hannah the only possible value of $v, so $v is deducible. Only $w remains as a relevant, non-deducible variable. To solve the problem, $w must thus be queried by the Agent.

Example 2 shows a problem that requires two queries to solve. Here all the variables are relevant and none are deducible. Querying, say, $v yields $v = Joe. At this point, $w becomes deducible and we can infer $w = Joe. We now know that Joe is either in the patio or in the basement but not which. A further query for $x reveals that $x = Hannah, so the answer is that Joe is in the patio.

We can visualize the possible solutions to these problems using the "query graphs"

C1. Hannah is in the garden.
C2. $u is Emma.
C3. $u is in the garden.
C4. The gift is in the garden.
C5. John is in the kitchen.
C6. The ball is in the kitchen.
C7. The skateboard is in the kitchen
E1. Hannah picks up the gift.
E2. John picks up $x.
E3. $v goes from the garden to the kitchen.
E4. $w walks from the kitchen to the patio.
E5. Having left the garden, $u goes to the patio.
Q. Where is the gift?
GT. $v = Hannah; $w = Hannah; Answer = Patio

Example 1: A QRAQ Problem

C1. Joe is in the kitchen.
C2. Bob is in the kitchen.
C3. Hannah is in the patio.
E1. $v goes from the kitchen to the garden.
E2. $w goes from the garden to the patio.
E3. $x goes from the patio to the basement.
Q. Where is Joe?
GT. $v = Joe; $w = Joe; $x = Hannah; answer = Patio

Example 2: A "deep" QRAQ problem (see Figure 1 for explanation)

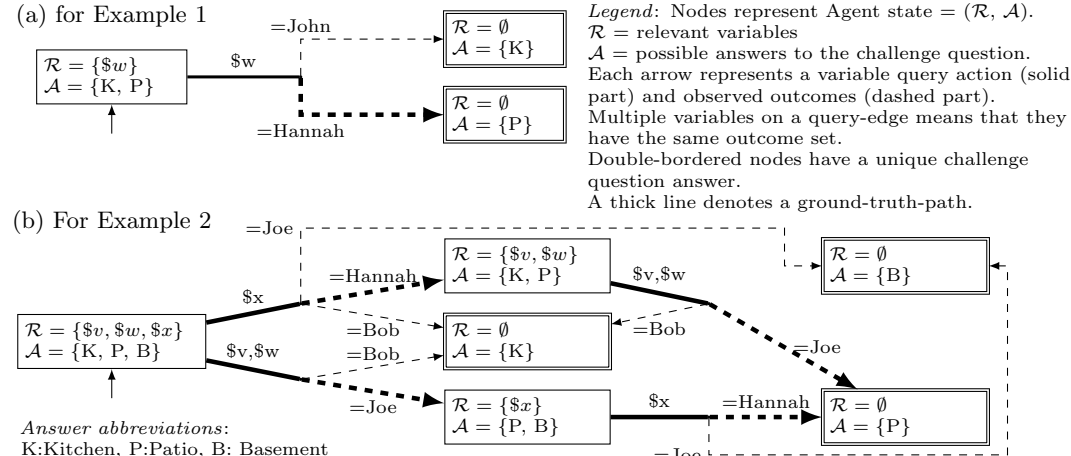

Answer abbreviations:
K:Kitchen, P:Patio, B: Basement

Figure 1: The QRAQ query graphs for both examples, illustrating the depth parameter

shown in Figure 1. These graphs (technically DAGs) show the query policy required to solve the problem. Each node represents an informational state of the agent, and each edge the outcome of relevant queries. We define the *depth* of a query graph to be maximum number of variables that must be queried in the worst case[2], given the ground truth variable assignments. Examples of such paths in the graphs of Figure 1 are shown in bold. These paths pass only through states consistent with the ground truth. By this definition, Figure 1(a) has depth 1, and figure 1(b) has depth 2. As we discuss in Section 5.2, depth is an important driver of problem complexity.

There are many ways to solve these problems algorithmically. The QRAQ simulator uses a linear-time, graph-based approach to track the possible locations of the protagonist as it processes the events. We will include a detailed description of the simulator and this algorithm when we release the QRAQ datasets to the research community.

## 4 Learning Agents

We develop and compare two different RL agents: (1) baseRL, which uses the memory network architecture from Sukhbaatar et al. (2015), and (2) impRL, which improves on the memory network architecture of baseRL by using a soft-attention mechanism over memory hops. We emphasize that the only feedback available to both agents, apart from a per-step penalty, is whether their answer to the challenge question is correct or not.

### 4.1 Control Loop

Both baseRL and impRL use the same control loop shown in Figure 2(a) whose steps are explained below:

**(1) Initialization.** For each problem, the challenge question is represented as a vector $c$, where $c_i$ is the index of the $i$-th word in the question in a dictionary. The events and context sentences are encoded similarly and then the event vectors are appended to the context vectors to construct an initial memory matrix $S_1$, where $S_1^{ij}$ is the index of the $j$-th word in the $i$-th sentence in the dictionary. Each word or variable comes from a global dictionary of $N$ words formed by the union of all the words in the training and testing problems.

**(2) Action Selection.** The agent's policy at the $t^{th}$ turn in the dialog is a function $\pi(a|S_t, c)$ mapping the memory matrix at turn $t$, $S_t$, and question, $c$, into a distribution over actions. An action, $a$, could be either a query for a variable or a final answer to the challenge question. The policy network, based on the end-to-end memory network, is shown in section 4.2.

**(3) Variable Query and Memory Update.** Whenever a query action is selected, the user module provides the true value for the corresponding variable in action $a_t$, i.e., the user module provides $v_t = \text{oracle}(a_t)$, where $v_t$ is the dictionary index of the true word for the variable in action $a_t$. Then all occurrences of variable in action $a_t$ in the memory $S_t$ are replaced with the true value $v_t$: $S_{t+1} = S_t[a_t \to v_t]$. The new memory representation $S_{t+1}$ is then used to determine the next action $a_{t+1}$ in the next turn of the dialog.

**(4) Final Answer Generation and Termination.** If the action is an answer instead of a variable to query, the task terminates and a reward is generated based on whether the answer is correct or not (for the exact values of the rewards, see paragraph on "curriculum learning" in section 5).

### 4.2 baseRL: End-to-End Memory Network based Policy Learner

The baseRL agent builds on an end-to-end memory network policy as introduced by Sukhbaatar et al. (2015). It maps the memory matrix, $S$, and the challenge question representation, $c$, into an action distribution. Specifically, the $i$-th row of the memory matrix is encoded into a vector $m_i = \sum_j l_j \circ A[S^{ij}]$, where $A \in \mathbb{R}^{d \times N}$ is an embedding matrix and $A[k]$ returns the $k$-th column vector, $d$ is the dimension of the embedding, '$\circ$' is an element wise multiplication operator, and $l_j$ is a column vector with the structure $l_j^k = (1 - j/J) - (k/d)(1 - 2j/J)$ with $J$ being the number of words in the sentences. Similarly, the challenge question is converted into a vector $q = \sum_j l_j \circ A[c_j]$.

---

[2]In the best case the agent may get lucky and query a variable that yields the answer immediately.

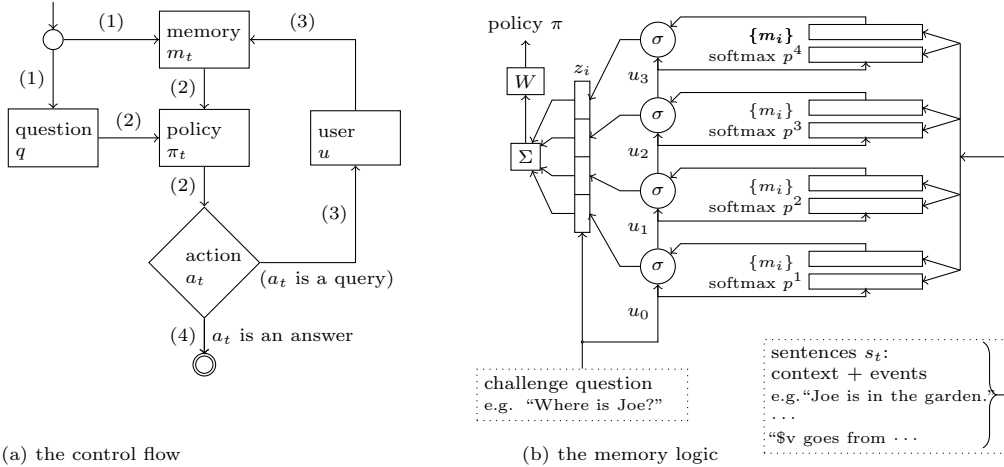

(a) the control flow
(b) the memory logic

Figure 2: The architecture. Figure (a) represents the control flow of the interaction of both our RL architectures with the User. Figure (b) shows the impRL memory network architecture. See text for details.

The output vector upon addressing and then reading from $\{m_i\}$ in the $k$-th hop is $u_k$:

$$u_k = \tanh(H(o_k + u_{k-1})) \text{ where } \qquad o_k = \sum_i p_i^k m_i, \qquad p_i^k = \text{SoftMax}(u_{k-1}^\intercal m_i), \qquad (1)$$

Where $\text{SoftMax}(z_i) = e^{z_i}/\sum_j e^{z_j}$ and $u_0 = q$. The policy module is implemented by two separate networks. One is for querying variables and the other is for answering the challenge questions.

**Query network output** The output of the query network is a distribution over variables in the memory matrix[3]. Since the problems have at most one variable per sentence, the distribution can be converted into the distribution over sentences without adding additional parameters. Specifically, baseRL implements the following query policy:

$$\pi_Q^i = \text{SoftMax}(u_M^\intercal m_i) \qquad (2)$$

where $u_M$ is the output of the last memory hop and only sentences with variables are considered in the SoftMax computation.

**Answer network output** The output of the policy module is a distribution over potential actions $\{a_1, ..., a_K\}$. Specifically, baseRL implements the following policy:

$$\pi_A = \text{SoftMax}(Wu_M + b) \qquad (3)$$

where $u_M$ is the output of the last memory hop, $W \in \mathbb{R}^{K \times d}$ and $b \in \mathbb{R}^K$.

### 4.3 IMPRL: IMPROVED END-TO-END MEMORY NETWORK BASED POLICY LEARNER

In preliminary experiments with the baseRL architecture we tried optimizing the number of memory hops and discovered that the optimal number varied with each dataset. Rather than try to optimize it as a hyperparameter for each dataset, we developed a new architecture shown in Figure 2(b). Unlike the baseRL agent in which the final action-distribution output only conditions on the last hop output from the memory network architecture, the improved RL architecture, impRL, computes the final action-distribution-output over all memory hop outputs (with a fixed number of total hops across all datasets) as follows:

$$\pi_Q^i = \text{SoftMax}(u^\intercal m_i) \quad \pi_A = \text{SoftMax}(Wu + b) \quad u = \sum_j z_j u_j \quad z_j = \text{SoftMax}(q^\intercal u_j) \quad (4)$$

---

[3]A special query action is added to signal a switch from the query network to the answer network.

where $W \in \mathbb{R}^{K \times d}$ and $b \in \mathbb{R}^K$. Our modification to the memory network architecture of Sukhbaatar et al. (2015) is similar to the adaptive memory mechanism (AM) employed by Weston et al. (2016). The AM mechanism allows a variable number of memory hop computations in memory networks, but it is trained via extra supervision signals to determine when to stop memory hop computation explicitly. Our impRL mechanism departs from the AM mechanism in two ways: (1) it does not require extra supervision signals, and (2) while the AM mechanism uses the last memory hop's output for prediction, impRL instead uses a weighted average of all memory hops' outputs. As we show below impRL improves performance over baseRL in our empirical results for the more challenging datasets.

## 4.4 Policy Gradient & Reward Function

Much of the prior work on reasoning problems has used supervised learning (SL) methods. Here we focus on the far more realistic and challenging setting where the learning agent only has access to a binary reward function that evaluates whether the answer provided by the agent to the challenge question is correct or not. More specifically, we use a reward function that is positive when the action is the correct answer to the challenge question, and negative when the action is a wrong answer to the challenge question or the action is a query to a variable (see section 5 Curriculum Learning for the details). The penalty for a wrong answer is much larger than the penalty for querying a variable. Penalizing queries encourages the agent to query for the value of as few variables as possible. The objective function of the reinforcement learning method is to optimize the expected cumulative reward over (say $M$) training problem instances: $\sum_{m=1}^{M} \mathbb{E}\{\sum_t r_t^m\}$, where $r_t^m$ is the immediate reward at the time step $t$ of the $m$-th task. The GPOMDP algorithm with average reward baseline (Weaver & Tao (2001)) is used to calculate the policy gradient which in turn becomes the error signal to train all the parameters in both baseRL and impRL.

## 5 Data, Training, and Results

We evaluate our methods on four types of datasets described below. Each dataset contains 107,000 QRAQ problems, with 100,000 for training, 2000 for testing, and 5000 for validation.

**(Loc)**. In this dataset, the context sentences describe people in rooms. The event sentences describe people (either by their name or by a variable) moving from room to room. The challenge questions are about the location of a specific person. We created several such datasets, scaling along vocabulary size, the number of sentences per problem (including both the context and event sentences), the number of variables per problem and the depth of the problem. For a definition of "depth" see Section 3. For a detailed configuration of each data set see Table 1.

**(+obj)**. This dataset adds objects to the Loc dataset. The context sentences describe people and objects in rooms. The event sentences describe people moving to and from various rooms, or picking up or dropping objects in rooms. People or objects (but not rooms) may be hidden by variables. The challenge questions are about the location of a person or an object.

**(+alias)**. Adds aliases to the Loc dataset. Some of them are defined in the context.

**(+par)**. This dataset modifies the Loc dataset by substituting sentences with semantically equivalent paraphrases. These paraphrases can change the number of words per sentence, and the ordering of the subject and object.

**Pre-training**   The embedding matrix $A$ is pre-trained via sentence level self-reconstruction. In the memory matrix $S$, each sentence $i$ is represented as a row vector $S^i$ and its corresponding embedding vector is $m_i = \sum_j l_j \cdot A[S^{ij}]$. The self-reconstruction objective is to maximize $\sum_i \sum_j \log(p_j(S^{ij}|m_i))$, where $p_j = \text{SoftMax}(W^{(j)}m_i)$ and $W \in \mathbb{R}^{N \times d}$. If a word position $j$ has only one word candidate, then such a position is dropped from the objective function.

Table 1: Datasets. The first 7 rows give statistics on the datasets themselves. The last 8 rows show results for answer accuracy (AnsAcc), trajectory accuracy (TrajAcc), trajectory completeness (TrajCmpl) and query accuracy (QryAcc) for the impRL and baseRL agents on the respective datasets. The middle 8 rows show results for the supervised learning agents.

| *Data Set* | *(Loc)* | | | | | *(+obj)* | *(+alias)* | *(+par)* |
|---|---|---|---|---|---|---|---|---|
| #names in vocab | 5 | 20 | 10 | 20 | 20 | 20 | 20 | 20 |
| #var in vocab | 5 | 20 | 10 | 20 | 20 | 20 | 20 | 20 |
| #sentence/prob. | 5-6 | 5-6 | 7-10 | 15-20 | 19-23 | 7-10 | 10-12 | 10-12 |
| #var/prob. | 0-2 | 0-2 | 0-2 | 0-3 | 5-10 | 5-10 | 0-5 | 0 |
| depth | 0-2 | 0-2 | 0-2 | 0-2 | 4-9 | 0-2 | 0-5 | 0 |
| avg. depth | 0.817 | 0.872 | 0.558 | 0.459 | 5.087 | 0.543 | 1.066 | - |
| sum(depth) / sum(#var) | 0.734 | 0.748 | 0.313 | 0.204 | 0.703 | 0.404 | 0.310 | - |
| AnsAcc in %; impSL | 99.9 | 99.5 | 92.1 | 95.3 | 91.4 | 95.9 | 90.7 | 99.8 |
| AnsAcc in %; baseSL | 99.9 | 99.2 | 92.3 | 92.4 | 90.2 | 95.5 | 86.6 | 98.8 |
| TrajAcc in %; impSL | 99.6 | 98.9 | 90.2 | 88.4 | 85.3 | 95.2 | 86.7 | - |
| TrajAcc in %; baseSL | 98.9 | 98.7 | 90.3 | 86.5 | 83.3 | 94.9 | 85.3 | - |
| TrajCmpl in %; impSL | 99.5 | 98.8 | 89.9 | 85.6 | 80.9 | 94.9 | 83.6 | - |
| TrajCmpl in %; baseSL | 98.7 | 98.7 | 90.0 | 83.5 | 78.7 | 94.6 | 82.9 | - |
| QryAcc in %; impSL | 99.5 | 99.2 | 96.4 | 84.6 | 93.5 | 97.7 | 93.7 | - |
| QryAcc in %; baseSL | 98.7 | 99.3 | 96.3 | 85.5 | 92.7 | 97.5 | 97.0 | - |
| AnsAcc in %; impRL | 99.1 | 94.4 | 86.5 | 89.0 | 64.2 | 81.1 | 75.7 | 96.9 |
| AnsAcc in %; baseRL | 98.4 | 95.0 | 88.4 | 88.2 | 54.6 | 79.6 | 69.7 | 97.2 |
| TrajAcc in %; impRL | 94.5 | 90.9 | 61.9 | 52.0 | 45.1 | 74.9 | 63.2 | - |
| TrajAcc in %; baseRL | 94.8 | 90.4 | 63.6 | 52.5 | 35.7 | 73.9 | 60.5 | - |
| TrajCmpl in %; impRL | 94.5 | 88.7 | 55.8 | 46.9 | 37.8 | 61.8 | 56.4 | - |
| TrajCmpl in %; baseRL | 94.6 | 89.5 | 59.9 | 47.4 | 28.3 | 61.2 | 54.5 | - |
| QryAcc in %; impRL | 94.3 | 95.4 | 49.2 | 32.1 | 80.0 | 69.6 | 77.0 | - |
| QryAcc in %; baseRL | 95.5 | 94.1 | 54.6 | 32.0 | 76.5 | 71.0 | 79.6 | - |

**Time embedding for events** If the $i$-th sentence is an event, a temporal embedding is added to its original embedding so that $m_i \leftarrow m_i + T(i)$, where $T(i) \in \mathbb{R}^d$ is the $i$-th column of a special matrix $T$ that is learned from data.

**Curriculum Learning** We first encourage both reinforcement learning agents to query variables by assigning positive rewards for querying any variable. After convergence under this initial reward function, we switch to the true reward function that assigns a negative reward for querying a variable to reduce the number of unnecessary queries. Specifically, the rewards is +1 for correct final answers, -5 for wrong final answers. We explored five pairs of query reward values for the curriculum: +/-0.01, +/-0.05, +/-0.1, +/-0.5, +/-1, and found that +/-0.05 performed best on a validation set, so that is what we use for our experiments.

**Action Mask** When choosing an action (query-variable or answer) from the policy network, output units that correspond to choices *not* available in the set of sentences at that turn[4] in the dialog are not considered in the selection process.

**Exploration** To encourage exploration, which is needed in RL settings, an exploration policy $\pi'$ is defined, given the agent's learned policy $\pi$ as follows. A random action is chosen with probability $\epsilon$ and the remaining $1 - \epsilon$ probability is distributed over actions as $\pi'(a) = (\pi(a) + \delta)/(1 + |A|\delta)$, where $\delta = \epsilon/(1 - \epsilon)|A|$, and $|A|$ is the number of actions. For our experiments, $\epsilon = 0.1$.

We used Adam (Kingma & Ba (2015)) to optimize the parameters of the networks. The number of memory hops is fixed to 4. The embedding dimensionality is fixed to 50.

---

[4]When a variable is queried, the simulation engine replaces every occurrence of a variable with its value and returns the updated text to the agent, at the beginning of the next turn.

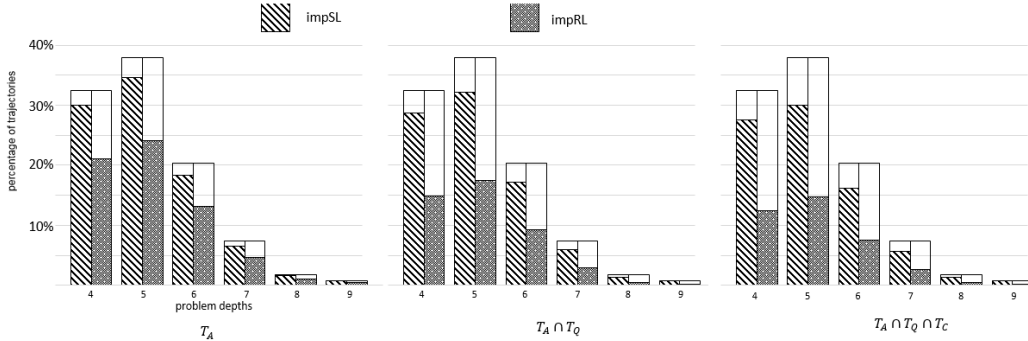

Figure 3: Test trajectory distributions over problem depths for impSL and impRL on the hardest Loc dataset. The bars show the the percentage of trajectories satisfying corresponding criteria for impSL and impRL. The figures show that impSL performs much better than impRL on deeper problems.

## 5.1 SUPERVISED LEARNING BASELINES: baseSL AND impSL

We trained the architecture of baseRL and impRL using supervised learning to get an upper-bound on achievable reinforcement learning performance. In the supervised learning setting, the relevant variables and (when appropriate) the correct final answers are provided at each turn, and the cross entropy loss function is used to optimize the parameters.

## 5.2 RESULTS & ANALYSIS

In our experiments we measure four metrics for each QRAQ datatset: (1) *answer-accuracy*, the proportion of trajectories (a sequence of variable queries followed by an answer) in which the agent answered correctly, but may have queried irrelevant variables or failed to query relevant ones; (2) *trajectory-accuracy*, the proportion of trajectories in which the agent queried only relevant variables then answered correctly, but may have left some relevant variables unqueried; (3) *trajectory-completeness*, the proportion of trajectories in which the agent queried all and only relevant variables before answering correctly; and (4) *query accuracy*, the proportion of correct queries among all queries made in any trajectory. Note that *trajectory-completeness <= trajectory-accuracy <= answer-accuracy* and *trajectory-completeness* is the most challenging metric by which we measure the success of our experiments.

More precisely, let $\mathcal{T}$ be the set of all trajectories produced by the agent on the test problems, $\mathcal{T}_A$ be the set of trajectories where the agent gave the correct answer, $\mathcal{T}_Q$ be the set of trajectories completed with only relevant variables queried, $\mathcal{T}_C$ be the set of trajectories completed with all relevant variables queried, $\mathcal{N}_Q$ be the number of queried variables, and $\mathcal{N}_{relQ}$ be the number of queried variables that were relevant. Then

$$\text{answer-accuracy} = \frac{|\mathcal{T}_A|}{|\mathcal{T}|}; \qquad \text{trajectory-completeness} = \frac{|\mathcal{T}_A \cap \mathcal{T}_Q \cap \mathcal{T}_C|}{|\mathcal{T}|};$$

$$\text{trajectory-accuracy} = \frac{|\mathcal{T}_A \cap \mathcal{T}_Q|}{|\mathcal{T}|}; \qquad \text{query-accuracy} = \frac{\mathcal{N}_{relQ}}{\mathcal{N}_Q}.$$

By comparing the various metrics in the rows of Table 1 we can make a number interesting observations. First, the performance gap between the supervised learning agents and reinforcement learning agents increases as problems become more complex. Figure 3 shows the test trajectory distributions over problem depths. The result shows that impSL performs much better than impRL on deeper problems. This indicates that the RL agent is sensitive to the scaling and hasn't learned as robust an algorithm as the supervised agent (which, considering the stronger training signal for the supervised agent, was to be expected). Second, for all reinforcement learning agents, the answer accuracy is considerably higher than the trajectory accuracy. This indicates that answering is considerably easier for the agent than solving the whole problem perfectly, with only relevant queries followed by a correct answer.

Third, in the simplest (Loc) datasets (the leftmost 3 to 4 columns) where the number of sentences, number of variables, and depth are all small, both baseRL and impRL do well and furthermore do similarly well. This is also the case for the +obj dataset because it is also similarly simple (in terms of depth range). As expected the answer accuracy, the trajectory accuracy and trajectory completeness get worse as the problems get more complex (in terms of parameters listed in the first 7 rows). This decrease in performance is roughly seen left to right in the 5 columns for the (Loc) datasets. Next, note that the query accuracy results of the leftmost 5 columns closely track the ratio of depth to the number of variables in the problems. That ratio is a rough estimate of the percentage of relevant variables to the total number of variables. An agent who guessed which variables were relevant would be sensitive to this ratio, doing better when it was high and worse when low, lending weight to the hypothesis that the RL agent's query algorithm is underperforming. Of all the parameters explored, the 'depth' and '#sentence/prob' parameters seem the most impactful. Specifically there is a sharp drop-off in performance for both the base and improved architectures in the rightmost (Loc) column where the depth is 4-9 compared with 0-2 in the leftmost four (Loc) columns. Similarly the (+obj) dataset has a low depth and the performance is similarly good. Finally, we note that in the more complex data sets including the rightmost column of the (Loc) set of columns as well as for the (+alias) dataset the performance of both baseRL and impRL is worse than for the simpler datasets but in all these cases impRL improves the answer accuracy significantly.

## 6 CONCLUSION

We have introduced the new QRAQ domain that presents an agent with a story consisting of context sentences, temporally ordered event sentences with variables, and a challenge question. Answering the challenge question requires multi-turn interactions in which a good agent should ask only non-deducible and relevant questions at any turn. We presented and evaluated two RL-based memory network architectures, a baseline and an improved architecture in which we added soft-attention over multiple memory hops. Our results show that that both architectures do solve these challenging problems to substantial degrees, even on the quite unforgiving trajectory-completeness measure, and despite being limited to feedback only about whether the the answer is correct. At the same time, as the gap between the supervised and RL approaches shows, there is considerable room for innovation in the reinforcement learning setting for this domain.

ACKNOWLEDGMENTS

Xiaoxiao Guo and Satinder Singh's work on this paper was supported by funding from IBM's Cognitive Horizons Network. Clemens Rosenbaum's work was conducted while an intern at IBM Research. The authors would like to thank Iulian Serban for helpful conversations.

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
