# Peer review of "Learning to Query, Reason, and Answer Questions On Ambiguous Texts"

_ICLR 2017 — accepted_

[Public Comment · Tara N Sainath · 07 Nov 2016]
**ICLR Paper Format**

Dear Authors,

Please resubmit your paper in the ICLR 2017 format with the correct font for your submission to be considered. Thank you!

[Official Review · AnonReviewer3 · rating 7 · confidence 3 · 16 Dec 2016]

This paper investigates a set of tasks that augment the basic bAbI problems. In particular, some of the people and objects in the scenarios are replaced with unknown variables. Some of these variables must be known to solve the question, thus the agent must learn to query for the values of these variables. Interestingly, one can now measure both the performance of the agent in correctly answering the question, and its efficiency in asking for the values of the correct unknown variables (and not variables that are unnecessary to answer the question). This inferring of unknown variables goes beyond what is required for the vanilla version of the bAbI tasks, which are now more or less solved.

The paper is well-written, and the contributions are clear. Due to the very limited vocabulary and structure of the bAbI problems in general, I think these tasks (and variants on them) should be viewed more as basic reasoning tasks than natural language understanding. I’m not convinced by the claim of the paper that this really tests the ‘interaction’ capabilities of agents – while the task is phrased as a kind of interaction, I think it’s more aptly described by simply ‘inferring important unknown variables’, which (while important) is more related to reasoning. I’m not sure whether the connection of this ability to ‘interaction’ is more a superficial one.

That being said, it is certainly true that conversational agents will need basic reasoning abilities to converse meaningfully with humans. I sympathise with the general goal of the bAbI tasks, which is to test these reasoning abilities in synthetic environments, that are just complicated enough (but not more) to drive the construction of interesting models. I am convinced by the authors that their extension to these tasks are interesting and worthy of future investigation, and thus I recommend the acceptance of the paper.

[Official Review · AnonReviewer1 · rating 7 · confidence 4 · 17 Dec 2016]
**No Title**

This paper introduces a nice dataset/data generator that creates bAbI like tasks, except where the questioning answering agent is required to clarify the values of some variables in order to succeed.  I think the baselines the authors use to test the tasks are appropriate.   I am a bit worried that the tasks may be too easy (as the bAbI tasks have been), but still, I think locally these will be useful.  If the generation code is well written, and the tasks are extensible, they may be useful for some time.

[Official Review · AnonReviewer2 · rating 6 · confidence 3 · 20 Dec 2016]
**No Title**

This paper proposed an integration of memory network with reinforcement learning. The experimental data is simple, but the model is very interesting and relatively novel. There are some questions about the model:

1. how does the model extend to the case with multiple variables in a single sentence?

2. If the answer is out of vocabulary, how would the model handle it?

3. I hope the authors can provide more analysis about the curriculum learning part, since it is very important for the RL model training.

4. In the training, in each iteration, how the data samples were selected, by random or from simple one depth to multiple depth?

[Author Response · Tim Klinger · 27 Dec 2016]
**Thanks to all the reviewers!**

Happy holidays and thank you all for your careful reading and thoughtful, constructive comments!

A few responses to your last set of comments/questions:

To Reviewer 3:

We definitely agree that because of the limited vocabulary and grammar, this is more about reasoning than language.  As you point out, the interaction in these problems  boils down to repeated variable queries until the Agent decides it has enough info.  We decided on the term "interaction" to differentiate our problems from the bAbI problems, which are 1 shot question answering, and to emphasize the idea that, although simplified, this has the same structure as a dialogue interaction with the Agent (for example a tech support agent) asking clarifying questions of the User until being able to answer the User's question. We can try to think of a better term -- perhaps "multi-turn"?

To Reviewer 2:

(1) how does the model extend to the case with multiple variables in a single sentence?

Currently we leverage the fact that there is just one variable per sentence to save on parameters and avoid a linear transformation before the softmax in the query network.  To support more than one variable per sentence would just require adding a linear transformation on the output before softmax, making the query network have the same structure as the answer network.

(2) If the answer is out of vocabulary, how would the model handle it?

Out of vocabulary words are in general a problem for networks which do a softmax over a fixed vocabulary.  One potential way to address this would be to use something like the Pointer Networks of Vinyals, Fortunato, and Jaitly(

[Final Decision · Program Chairs · 06 Feb 2017]
**ICLR committee final decision**

The program committee appreciates the authors' response to concerns raised in the reviews. While there are some concerns with the paper that the authors are strongly encouraged to address for the final version of the paper, overall, the work has contributions that are worth presenting at ICLR.